# Impacts of the Expected Credit Loss Model on Pro-Cyclicality, Earnings Management, and Equity Management in the Portuguese Banking Sector

Miguel Resende [ID], Carla Carvalho [ID] and Cecília Carmo *[ID]

GOVCOPP—Research Unit on Governance, Competitiveness and Public Policies, Higher Institute of Accounting and Administration, University of Aveiro, 3810-193 Aveiro, Portugal; miguelresende@ua.pt (M.R.); carla.carvalho@ua.pt (C.C.)
* Correspondence: cecilia.carmo@ua.pt

**Abstract:** This article delves into the pro-cyclicality of loan loss provisions (LLPs) and earnings management, along with equity management, in Portuguese banks against the backdrop of implementing the IFRS 9's expected credit loss (ECL) model. It concentrates on how LLPs mirror economic cycles and financial management practices, providing valuable insights into the operational dynamics of the Portuguese banking sector, marked by distinct economic and regulatory challenges. The research examined a sample of five Portuguese commercial banks, chosen from a group of seventeen in the Portuguese Banking Association. Data spanning from 2013 to 2022 were manually gathered. A multiple linear regression model was employed to scrutinize the relationship between LLPs and variables indicative of economic cycles and the earnings and equity management. The methodology use was a multiple linear regression model. The analysis indicates a pro-cyclicality in LLPs within the Portuguese context, with a positive response of LLPs to economic indicators like unemployment. Contrarily, the extent of earnings and equity management under the ECL model was less marked compared to the incurred credit loss (ICL) model, suggesting the impact of more stringent regulatory measures. The research corroborates the pro-cyclicality of LLPs in Portuguese banks under the ECL framework, underscoring the necessity for ongoing monitoring and refinement of models for forecasting and recognizing credit losses. The findings point to an area for improvement in financial management practices, despite regulatory enhancements, to promote transparency and ensure financial stability.

**Keywords:** earnings management; equity management; pro-cyclicality; loan loss provision; IFRS 9; expected credit loss

## 1. Introduction

The subprime mortgage crisis, which sparked the 2007 financial crisis, exposed issues in accounting standards that contributed to the loss of confidence in the banking system. One of the primary weaknesses was the insufficient and delayed recognition of LLPs[1] arising from the diminished recoverable value of loans and other financial instruments (Bischof et al. 2021). Consequently, the need to modify the credit loss model emerged from the 2007 financial crisis, which precipitated economic crises worldwide, prompting a reevaluation of the approach to financial instruments (Pucci 2017). In response, the International Accounting Standards Board (IASB) published the International Financial Reporting Standard 9 (IFRS 9) in July 2014, replacing the International Accounting Standard 39 (IAS 39). IFRS 9 was developed based on a logical model for recognizing and measuring financial instruments, introducing the concept of expected losses (Silva 2017).

The existing model, supported by IAS 39 and known as the ICL, delayed the recognition of LLPs until there were clear and objective indications of impairment (Bushman and Williams 2015; Casta et al. 2019; Gebhardt and Novotny-Farkas 2011; Laux 2012). IAS 39

was formulated by the IASB to restrict companies' ability to constitute impairments without justification, using them as a means of earnings management[2] (Beck and Narayanamoorthy 2013; Camfferman 2015; Wall and Koch 2000).

As Harrald and Sandall (2010) point out, discussions about banking regulation aim to ensure the stability of the financial system, considering that the system can amplify the effects of economic expansion and contraction phases. Pro-cyclicality, or the positive correlation between a variable and economic activity, is at the heart of this debate (Bebczuk et al. 2011). For Longbrake and Rossi (2011), the financial system should, instead of amplifying, smooth out economic cycles. Pro-cyclicality can become particularly problematic by exacerbating the impact of a downturn in the economic cycle, exacerbating crises, and compromising stability. It is observed that at the beginning of a declining economic cycle, banks recognize few LLPs. However, as the crisis deepens, LLPs increase, which harms the banks' financial situation, reducing credit availability precisely when it is most needed in the market (Araújo et al. 2018). Berger and Udell (2004) and Bouvatier and Lepetit (2012) warn that the excessive cyclicality of bank credit can intensify the economic cycle, increase systemic risk, and harm the proper allocation of available resources for loans, highlighting the importance of mitigating the pro-cyclicality of the financial system to achieve economic stability.

In addition to economic cycles, earnings management in dynamic models is also a concern for standard setters, as well as the transparency of financial statements (FASB-IASB 2009). Compared to the ICL model of IAS 39, the IFRS 9's ECL model exhibits greater subjectivity, particularly in the judgment required for recognizing LLPs (Novotny-Farkas 2016). On the other hand, the ECL model can contribute to greater banking transparency and more effective market discipline, being fundamental in enhancing financial stability (Onali et al. 2021). Salazar et al. (2023) also confirm the improvement in transparency with IFRS 9 compared to IAS 39, considering the disclosed information on LLP and the new disclosures of phase 2 loan loss allowances (LLAs), providing stakeholders with more information on the anticipation of future risks.

In the evolving landscape of financial reporting and risk management, the implementation of the ECL model under IFRS 9 represents a paradigm shift towards more transparent and timely recognition of LLPs. This study, set against the backdrop of Portugal's unique economic and financial challenges within the Euro-zone—especially following the 2011 international intervention and the subsequent public recapitalization of key banks (Costa 2016)—aims to shed light on the significant implications of such regulatory changes. The decision to focus exclusively on Portugal provides a more uniform and controlled investigative framework, acknowledging the country's unique economic and market conditions. This targeted approach yields critical insights into the efficacy of recent accounting reforms, underscoring their role in bolstering financial sector resilience, transparency, and in reducing systemic risks. Ultimately, this comprehensive examination of the ECL model and related accounting standards like IFRS 9 establishes a foundation for future research in financial reporting and risk management, both within Portugal and in broader international contexts. This study aims to analyze the pro-cyclicality of LLPs in Portuguese banks, determining the existence of either a negative or positive relationship with economic cycles, and to assess the presence of earnings management and equity management through their recognition in five Portuguese commercial banks over the period from 2013 to 2022. The results indicate that there is pro-cyclicality in the ECL model, as well as earnings management and equity management. The methodology used to capture economic cycles, and earnings and equity management, is based on a multiple linear regression model adapted from the studies of Araújo et al. (2018), Beatty and Liao (2014), and Casta et al. (2019).

This research stands out from previous studies by focusing on the impact of LLPs recognition within a single jurisdiction, capturing only the local effects of the ECL model on Portuguese banks. The findings of this study will thus enable regulators and standard-setters to observe whether the ECL model is creating the necessary reserves to face periods of crisis and, simultaneously, if it is being used for earnings and equity management in

Portuguese banks. With this approach, the study makes a significant contribution to the literature on the behavior of LLPs recognition through the ECL model under IFRS 9, in economic cycles and its potential use for earnings and equity management.

The study is divided into six chapters. Following this introductory first chapter, there is a chapter on literature review and hypothesis formulation. The third chapter presents the analysis model and the variables used in the empirical study, as well as the sample and methods of data collection and processing. In the fourth chapter, the statistical analysis is carried out, and the results obtained are discussed. The fourth main point, Results, presents a comprehensive statistical analysis, scrutinizing the findings derived from the empirical investigation. Following this, the Discussion, designated as the fifth main point, critically examines the results, contextualizing them within the broader field of study. Finally, in the sixth chapter, the main conclusions of the study are presented along with its limitations and suggestions for future research are identified.

## 2. Literature Review
### 2.1. Cyclicality and Pro-Cyclicality of LLPs

Cyclicality refers to the cyclical nature of the economy, where periods of expansion and recession occur. These economic cycles can affect the quality of a financial institution's assets and, consequently, the LLPs (Longbrake and Rossi 2011). On the other hand, pro-cyclicality refers to a phenomenon where the policies, practices, or financial instruments adopted by financial institutions, or by regulators, amplify the cyclical fluctuations of the economy instead of smoothing them (Bebczuk et al. 2011). Borio et al. (2001) define pro-cyclicality as the tendency of the financial system to generate booms and financial collapses, more specifically considering the mechanisms that amplify these financial fluctuations. Ozili and Outa (2017), in their literature review study, mention that the term pro-cyclicality is used when banks enter a recession period, leading their managers to decrease credit granting and increase LLPs. This increase, during periods of recession, will further reduce the banks' net interest margin, decreasing profits and deteriorating their situation during the recession. Regarding LLPs, pro-cyclicality occurs when the actions of financial institutions amplify economic movements, increasing risks during periods of expansion and exacerbating recessions (Longbrake and Rossi 2011).

Pro-cyclicality has a positive effect during periods of economic growth (Longbrake and Rossi 2011) but contributes negatively at the turning point of the economic cycle, as well as during recessions, potentially delaying economic recovery. Furthermore, the actions in recognizing LLPs by financial institutions can also aggravate economic movements, increasing risks during recessions (Chen et al. 2020; Gomaa et al. 2019; Pastiranová and Witzany 2021). Some degree of pro-cyclicality is inevitable and inherent in economic activity, but the credit market intensifies the peaks and troughs of this cycle (Bikker and Metzemakers 2005; Borio et al. 2001).

Figure 1 aims to illustrate the effect of LLP pro-cyclicality on economic cycles as found in the literature (Donelian 2019; Novotny-Farkas 2016).

As depicted in Figure 1, at the onset of an economic downturn, banks recognize few LLPs. However, as the crisis worsens, LLPs increase, deteriorating banks' financial positions, reducing credit issuance (precisely when the market needs it most), intensifying the economic cycle, increasing systemic risk, and impairing the proper allocation of available loan resources (Berger and Udell 2004; Bouvatier and Lepetit 2012).

The literature on the pro-cyclicality of the credit market outlines both positive and negative aspects, as illuminated by various studies (Bouvatier and Lepetit 2012; Donelian 2019; Morrison and White 2010; Novotny-Farkas 2016; Ozili and Outa 2017; Pucci 2017). Positive pro-cyclicality occurs during economic upturns, where optimistic projections about future cash flows and macroeconomic factors lead banks to increase credit issuance, thus amplifying economic growth. However, this can turn into a vicious cycle, potentially leading to an unsustainable boom. Negative pro-cyclicality arises when economic expectations worsen, causing banks to reassess and increase credit risk, reduce credit issuance,

and thus exacerbate economic downturns, making recovery slow and challenging. This duality highlights the importance of managing and regulating credit practices to mitigate cyclical risks.

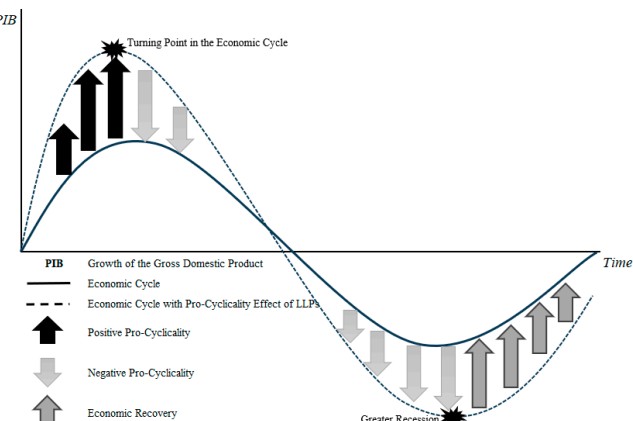

**Figure 1.** Economic Cycles and pro-cyclical effect of LLPs. Adapted from Novotny-Farkas (2016) and Donelian (2019).

The transition from ICL model under IAS 39 to the ECL model under IFRS 9 marks a pivotal shift in financial reporting and risk management, aiming to address the cyclicality and pro-cyclicality inherent in financial institutions' operations. The following analysis offers a comparative analysis of these models, particularly focusing on their impact on LLPs, cyclicality, and the broader financial ecosystem.

The ICL model, as operationalized within IAS 39, has been critiqued for its reactive nature, especially evident during the 2007 financial crisis. Researchers like Pucci (2017) and Novotny-Farkas (2016) highlight how the ICL model amplified economic cycles by failing to account for expected credit losses, leading to a reactive, rather than proactive, approach to financial instability. This delay in LLP recognition, underlined by the stringent criteria of IAS 39 (§§58–59), restricted banks from anticipating future losses, thus contributing to increased systemic risk and calling for a comprehensive overhaul of the credit loss recognition framework. The literature points out that the ICL model's lack of timeliness in recognizing LLPs contributed to increased systemic risk and financial instability (Bushman and Williams 2015; Gebhardt and Novotny-Farkas 2011; Hoogervorst 2014; Laux 2012), underscoring the need for a more forward-looking approach.

In response to these challenges, the IASB developed IFRS 9 (Ferreira 2011), introducing the ECL model as a means to incorporate a more predictive element into financial reporting. The ECL model represents a significant departure from the ICL model by requiring the recognition of credit losses based on expected, rather than incurred, losses. This change is predicated on the understanding that a proactive approach to recognizing LLPs can enhance financial stability by allowing institutions to buffer against potential future downturns (Casta et al. 2019). Despite its potential, the ECL model has encountered criticisms regarding its implementation challenges, notably the increase in credit risk assessment complexity and the potential for delayed LLP recognition (EBA 2021). For example, a conservative approach to credit risk might result in delays in recognizing LLPs. Therefore, a good alignment between regulators and standard-setters in defining default of credit losses is crucial to avoid divergences in the application of the ECL model by different institutions, as addressed by the IASB in its 2020 communication[3]. The methodology for calculating LLAs[4] under the ECL model, primarily based on probability of default (PD), loss given default (LGD), and exposure at default (EAD), reflects an attempt to quantify credit risk more accurately, yet its efficacy remains subject to financial institutions' ability to accurately forecast and apply these metrics (KPMG 2016; López-Espinosa et al. 2021; Volarević and Varović 2018).

As Novotny-Farkas (2016) notes, all LLPs accounting models that seek to reflect economic conditions are inherently pro-cyclical. In the same study, the author states that the approach of the IFRS 9's ECL model can mitigate concerns about pro-cyclicality. The recognition of 12-month LLPs for phase 1 serves, to some extent, as an adjustment for the credit spread recognized through income, thus resulting in less overstated profits. This will reduce the possibility of distributing overstated profits in the form of dividends and bonuses during periods of economic expansion and will result in more capital to support losses during a downturn. Moreover, the more timely recognition of LLPs and their impact on regulatory capital can mitigate excessive loan growth during periods of economic expansion. In fact, Novotny-Farkas (2016) state that in phase 1, overstated LLAs are expected, having a "buffer effect on regulatory" that increases with the risk of newly granted loans.

Novotny-Farkas (2016) believes that the timelier recognition of LLPs and a higher level of disclosures will promote market discipline. Specifically, providing timely information about credit risk (increases in credit risk) to market participants can reduce financing constraints in times of financial stress. Thus, with a dynamic model, such as the IFRS 9's ECL model, banks are expected to be less susceptible to the pro-cyclical effects of credit risk, as:

- At the peak of the economic cycle, PDs and LGDs are lower, requiring less capital, increasing regulatory capital, and allowing for more credit to be granted for the same amount of available capital. However, in the face of phase 1 of the ECL model, a containment in excessive credit granting is expected considering economic forecasts.
- At the onset of a potential recession period, the PD and LGD are higher, necessitating more capital, thus reducing regulatory capital and, consequently, the availability to grant more credit. It is expected that the LLAs will be larger, enabling the absorption of future LLPs and maintaining credit provision, contributing to the reversal of the downward economic cycle.

### 2.2. The Relationship between LLPs and Economic Cycles

Various authors have focused their studies on the behavior of LLPs during economic cycles, arguing that they are pro-cyclical, as they reinforce the current state of the economy (Agénor and Zilberman 2015; Bikker and Metzemakers 2005; Bouvatier and Lepetit 2012; Olszak et al. 2017). The recognition of LLPs tends to be substantially higher in a downward economic cycle, reflecting the growth of risk (Bikker and Metzemakers 2005). Bouvatier and Lepetit (2012) analyzed how the rules for recognizing LLPs influence the loan market, concluding that a credit loss system based on past information increases the pro-cyclicality of market fluctuations. According to the authors, in a credit loss system based on predictions of future losses, the problem of pro-cyclicality does not exist. Agénor and Zilberman (2015) also show that in a backward-looking credit loss model, LLPs are pro-cyclical because they are triggered by overdue payments (or non-performing loans), which depend on current economic conditions and the relationship between LLAs and loans. Olszak et al. (2017) found that LLPs in large, publicly traded commercial banks, as well as those in banks reporting consolidated financial statements, are more pro-cyclical. Conversely, they found that the existence of strict capital standards and better investor protection are associated with less pro-cyclicality in LLPs.

Ozili and Outa (2017) questioned to what extent dynamic norms should be applied differently in developed and emerging countries. In Ozili (2019)'s study, some clues are found, highlighting the role of regulators in the proper application of the standard. In that study, evidence was found that African banks in more corrupt environments tend to smooth their positive results more aggressively, with this smoothing being reduced among African banks in environments with strong investor protection. Thus, in jurisdictions where the environment and incentive for earnings management are greater, regulators will play a key role in the proper application of the standard (Marton and Runesson 2017). Saurina (2009) emphasizes the importance of regulators' and financial institutions' awareness of

pro-cyclicality and the need to adopt measures to mitigate its negative effects, aiming for proper credit risk management and reduction of LLPs, opting for a dynamic system for credit losses to mitigate pro-cyclicality. In 2020, due to the COVID-19 pandemic and its consequences on the real economy, European Banking Authority (EBA) moratorium guidelines were implemented to prevent volatile impairments and safeguard the quality of banking profits. During this period, banks faced significant challenges in determining significant increases in credit risk, which consequently had repercussions on the amounts of the different phases of the ECL model, limiting its impact on LLPs (Salazar et al. 2023). Krüger et al. (2018) also argue that in times of crisis, regulators have the possibility to mitigate the effects of recessions, either by decreasing cyclical buffers or by modifying the approach to LLPs, especially when banking institutions face adversities.

In the context of Portuguese banks, the implementation of IFRS 9 and its ECL model takes on particular significance given the country's recent experience with financial crises and subsequent interventions to stabilize the banking sector. Following the global financial crisis, Portugal faced significant economic challenges, culminating in a request for external intervention. During this period, Portuguese banks underwent comprehensive stress testing by the EBA in cooperation with the European Systemic Risk Board (ESRB) and Banco de Portugal aimed at assessing their resilience in the face of adverse scenarios (Banco de Portugal 2011). These tests focused on the country's major banking groups (CGD, BCP, BES, and BPI), revealing the sector's vulnerability to external shocks, exacerbated by the sovereign debt crisis in the Eurozone. The deterioration of banks' market capitalization, compounded by the loss of access to international debt markets and sovereign debt rating downgrades, highlighted the fragility of the Portuguese banking system (Banco de Portugal 2011). The response to these challenges included public recapitalization measures and initiatives to reduce the indebtedness of the non-financial sector, underscoring the importance of sustaining banks' business models and ensuring their solvency (Augusto and Félix 2014; Costa 2016). The role of regulators is crucial, especially in environments prone to earnings management, to prevent conservative loss accounting from becoming an incentive for pro-cyclicality, exacerbating rather than mitigating crises (Marton and Runesson 2017; Laux 2012).

The literature reviewed predicts that the cyclical nature of LLPs is common and that dynamic models attempt to mitigate this pro-cyclicality, recognizing that a certain pro-cyclicality is inherent in economic activity. It is important to prevent the credit market from amplifying the extremes of economic cycles (Agénor and Zilberman 2015; Bikker and Metzemakers 2005; Borio et al. 2001; Bouvatier and Lepetit 2012; Chen et al. 2020; Gomaa et al. 2019; Morrison and White 2010; Novotny-Farkas 2016; Olszak et al. 2017; Ozili and Outa 2017; Pastiranová and Witzany 2021; Pucci 2017). Based on this evidence, the first hypothesis to be tested is formulated:

**H1.** *The recognition of LLPs in Portuguese banks, according to the IFRS 9's ECL model, exhibits pro-cyclicality.*

*2.3. Impact of Dynamic Models on Economic Cycles Management*

The recent shift towards adopting the ECL model under IFRS 9 has sparked significant debate within the banking and regulatory communities, particularly concerning its potential to mitigate or exacerbate the pro-cyclicality of LLPs. This discussion is crucial in understanding how financial institutions can better prepare for economic fluctuations, ensuring stability and transparency in their operations. Regulators and standard-setters have been advocating for a more dynamic or anti-cyclical approach to credit loss accounting to counteract this trend (Ozili and Outa 2017). A dynamic credit loss system would entail banks reporting higher LLPs during economic upturns and recognizing lower LLPs during downturns. Such a system leverages LLAs accumulated in good times to cover credit losses in bad times, aiming to enhance bank safety and soundness by preventing insolvency during economic downturns (Balla and McKenna 2009; Saurina 2009).

However, the implementation of forward-looking models, including the ECL model under IFRS 9, has been met with criticism. Critics argue that these models can react strongly to economic changes, increasing the volatility of LLPs. For instance, Bikker and Metzemakers (2005) pointed out the challenge in detecting economic cycles, which complicates the application of dynamic models for LLPs. Studies have shown that the ECL model, while aiming to minimize risks through early recognition of LLPs, exhibits greater sensitivity to economic conditions compared to its predecessor, the ICL model under IAS 39 (Abad and Suarez 2017; Pastiranová and Witzany 2021, 2022; Chen et al. 2020; Engelmann and Nguyen 2023).

Before the advent of IFRS 9, some countries—like Spain, Peru, Uruguay, and Chile—had already implemented dynamic LLP systems. The transition from IAS 39 to IFRS 9 was expected to standardize LLP practices, but the application has been uneven, with some banks using high-cycle reserves to mask financial difficulties (Novotny-Farkas 2016; Gebhardt and Novotny-Farkas 2011). For example, the Bank of Spain encouraged Spanish banks to continue the previous regime, not reversing the LLAs, leading to an accumulation of past reserves alongside LLPs recognized under IAS 39. This process, incompatible with IAS 39 requirements, also made Spanish LLAs less transparent (Gebhardt and Novotny-Farkas 2011). Novotny-Farkas (2016) argued that these high-cycle reserves could be used by Spanish banks to conceal losses until their complete utilization, making banks appear healthy over the years when they were actually struggling financially. Eventually, several Spanish banks faced difficulties and required bailouts after depleting their LLAs and realizing hidden losses (Bloomberg 2012). Despite these challenges, there are arguments in favor of dynamic models. Studies in Chile and Uruguay suggest that while dynamic LLPs may not fully mitigate pro-cyclicality, they can enhance bank solvency and offset economic downturn impacts when paired with other measures like countercyclical capital buffers (Chan-Lau 2012; Wezel 2010).

Another aspect of concern is earnings management. The shift to the ECL model has increased the discretion bank managers have in recognizing LLPs, potentially allowing for more conservative or aggressive accounting practices. This could lead to greater earnings management, affecting the transparency and reliability of financial statements (Du et al. 2023; Gomaa et al. 2019; Novotny-Farkas 2016; Ozili and Outa 2017). Novotny-Farkas (2016) had already warned of greater subjectivity in this standard compared to IAS 39 in terms of judgment for recognizing LLPs. This new reality may allow bank administrations to manage earnings, as the ECL model is more forward-looking compared to the ICL model. Therefore, standard-setters' concerns might not be safeguarded, as the dynamic provisions present in this new model can potentially promote earnings management, impairing the transparency of financial statements (FASB-IASB 2009). Studies have shown mixed results regarding the impact of IFRS 9 on earnings management, with some indicating an increase in such practices, especially in less regulated environments (Nnadi et al. 2023; Norouzpour et al. 2023).

Given these findings, it is hypothesized that the recognition of LLPs under the ECL model of IFRS 9 allows for earnings and equity management in Portuguese banks. This hypothesis aligns with the broader literature indicating that the greater subjectivity of the ECL model compared to the ICL model increases discretionary power in LLP recognition, potentially facilitating earnings and equity management (Du et al. 2023; Gomaa et al. 2019; Novotny-Farkas 2016; Ozili and Outa 2017). Thus, the second hypothesis to be empirically tested is formulated:

**H2.** *The recognition of LLPs in Portuguese banks, according to the IFRS 9's ECL model, allows for earnings and equity management.*

## 3. Methodology

### 3.1. Analysis Model and Variables

To test the formulated hypotheses, the following multiple linear regression model was utilized, which describes the relationship between the dependent variable LLPs (explained variable) and the explanatory variables representative of economic cycles and the management of earnings and equity. This approach follows the methodology adopted in the studies of Araújo et al. (2018), Beatty and Liao (2014), and Casta et al. (2019).

$$LLP_{it} = \beta_0 + \beta_1 PIB_t + \beta_2 UNEMPLOYED_t + \beta_3 EBTP_{it} + \beta_4 EQUITY_{it} + \beta_5 \Delta LOANS_{it}$$
$$+ \beta_6 LOANS_{it} + \beta_7 LLA_{it} + \beta_8 SIZE_{it} + \beta_9 IFRS_t + \varepsilon_{it}$$

This study emphasizes the significance of the concepts of LLPs and LLAs for the analysis of credit impairments. Utilizing the approach of Salazar et al. (2023), LLPs represent the impairment losses recognized in a given period, while LLAs refer to the total accumulated anticipated losses on the balance sheet. The adoption of these terms seeks to ensure consistency with the literature and clarify the communication of the study's findings.

In Table 1, the description of the dependent, independent, and control variables used in the model is presented.

**Table 1.** Description of variables used in the model.

| Variable | Type | Definition | Previous Studies |
|---|---|---|---|
| $LLP_{it}$ | Dependent | Net LLPs over total assets of bank $i$ in year $t$ | Araújo et al. (2018), López-Espinosa et al. (2021), Nnadi et al. (2023), Norouzpour et al. (2023), and Pastiranová and Witzany (2022) |
| $PIB_t$ | Independent | Real GDP growth in year $t$ | Araújo et al. (2018), Beatty and Liao (2014), Casta et al. (2019), Marton and Runesson (2017), Nnadi et al. (2023), Norouzpour et al. (2023), Ozili and Outa (2017), and Pastiranová and Witzany (2022) |
| $UNEMPLOYED_t$ | Independent | Unemployment rate in year $t$ | Araújo et al. (2018), Beatty and Liao (2014), and Casta et al. (2019) |
| $EBTP_{it}$ | Independent | Earnings before taxes and LLPs over total assets of bank $i$ in year $t$ | Araújo et al. (2018), Nnadi et al. (2023), Norouzpour et al. (2023), and Ozili and Outa (2017) |
| $EQUITY_{it}$ | Independent | Equity over total assets of bank $i$ in year $t$ | Araújo et al. (2018), Casta et al. (2019), and Ozili and Outa (2017) |
| $\Delta LOANS_{it}$ | Control | Variation between total loans of year $t$ and year $t-1$, divided by total loans of year $t$ of bank $i$ | Araújo et al. (2018), Beatty and Liao (2014), Norouzpour et al. (2023), and Ozili and Outa (2017) |
| $LOANS_{it}$ | Control | Total loans over total assets of bank $i$ in year $t$ | Araújo et al. (2018), Ozili and Outa (2017), Marton and Runesson (2017), and Beatty and Liao (2014) |
| $LLA_{it}$ | Control | LLA in relation to total assets of bank $i$ in year $t$ | Beatty and Liao (2014), and Casta et al. (2019) |
| $SIZE_{it}$ | Control | Natural logarithm of total assets of bank $i$ in year $t$ | Araújo et al. (2018), Beatty and Liao (2014), Casta et al. (2019), Nnadi et al. (2023), and Norouzpour et al. (2023) |
| $IFRS_t$ | Control | Dummy variable that takes the value 1 in the years of application of IFRS 9, and the value 0 in the years of application of IAS 39 | Casta et al. (2019), Marton and Runesson (2017), Nnadi et al. (2023), and Norouzpour et al. (2023) |

Regarding the explained (dependent) variable, most models that address the relationship between the recognition of LLPs, economic cycles, and earnings management use the variable $LLP_{it}$ as the dependent variable (Araújo et al. 2018; López-Espinosa et al. 2021; Pastiranová and Witzany 2022).

As for the explanatory variables, the variables $PIB_t$ and $UNEMPLOYED_t$ are used to study the effects of economic cycles on the recognition of LLPs, and the variables $EBTP_{it}$ and $EQUITY_{it}$ are used to study earnings management and equity management in the recognition of LLPs. Control variables include bank size ($SIZE_{it}$), accounting standard used ($IFRS_t$), LLAs ($LLA_{it}$), and the amount and variation of loans granted ($LOANS_{it}$ e $\Delta LOANS_{it}$).

Table 2 succinctly presents the expected results regarding the behavior and sign of the coefficients associated with the explanatory and control variables, considering the formulated hypotheses. In other words, it summarizes the theoretical or expected predictions, based on the literature review, for the impact of independent and control variables on the dependent variable.

**Table 2.** Expected behavior of variables used in the model.

| Variable | Expected Behavior | Signal |
|:---:|:---|:---:|
| $PIB_t$ | Higher sensitivity to economic variations is expected in the ECL model, where a positive or negative change in GDP may decrease or increase, respectively, the level of LLPs, indicating the existence of cyclicality in the model. | $+/-$ |
| $UNEMPLOYED_t$ | Higher sensitivity to changes in economic measures is expected, where a positive or negative variation in unemployment rate may increase or decrease, respectively, the level of LLPs, signaling the existence of cyclicality in the model. | $+/-$ |
| $EBTP_{it}$ | Banks are expected to use LLPs as an earnings management tool, where an increase in earnings before taxes and LLPs will increase the level of LLPs. | $+$ |
| $EQUITY_{it}$ | Banks are expected to use LLPs as an equity management tool, where a negative variation in a bank's capital may increase the level of LLPs. Banks tend to increase LLPs when regulatory capital is below required levels. This practice is more common in banks with lower capitals. | $-$ |
| $\Delta LOANS_{it}$ | A positive variation in granted loans represents an increase in credit risk. Hence, a positive change in granted loans is expected to increase the value of LLPs. | $+$ |
| $LOANS_{it}$ | The larger the share of granted loans in a bank's total investments, the higher its credit risk. An increase in the ratio of total loans to total assets is expected to increase the value of LLPs. | $+$ |
| $LLA_{it}$ | The larger the LLAs in a bank's total investments, the higher its credit risk. An increase in the ratio of LLAs to total assets is expected to increase the value of LLPs. | $+$ |
| $SIZE_{it}$ | The larger the bank, the higher the levels of LLPs recognition. The larger the bank, the higher the value of LLPs is expected to be. | $+$ |
| $IFRS_t$ | Higher LLPs values are expected with the new ECL model of IFRS 9, compared to the ICL model of IAS 39. | $+$ |

To more thoroughly assess the impacts on LLPs and validate the two formulated hypotheses, the model will be estimated separately for two periods: from 2013 to 2017, corresponding to the application period of the ICL model; and from 2018 to 2022, corresponding to the application period of the ECL model. Dividing the sample into two distinct periods will allow for an analysis of the behavior of the explanatory variables in relation to the dependent variable, verifying the consistency of the main model's results. This detailed approach is essential to better understand the impacts of LLPs under both models (ICL and ECL) over the periods of their application.

### 3.2. Sample and Data

The sample for this study was defined using the database of the Portuguese Banking Association, which, as of April 2023, listed 17 banks with consolidated reports and accounts available for the semesters of 2012 to 2022, from which all necessary data were manually collected to construct the variables. In total, 12 banks were excluded from the study, specifically: BES, BESI, Banif, Sant Consumer, CBI, BBVA, Itaú, and Barclays for not being present throughout the entire study period and for having ceased to exist or no longer being part of the database; Invest and Crédito Agrícola due to structural changes during the analysis period; and BIG and Finantia due to the unavailability of all the necessary data for the study. Therefore, the final sample consists of 5 commercial banks. Despite the sample size being reduced to five banks, these institutions account for 68% of the total assets of the 17 banks represented in the database of the Portuguese Banking Association.

Table 3 presents the banks included in the sample, ordered by decreasing total assets as of 31 December 2022. As can be seen, on the last day of the year 2022, the total accumulated assets on the balance sheet of all the banks in the sample amounted to 309 billion euros.

**Table 3.** Sample characterization.

| Banks (in Thousands of Euros) | Total Assets 31 December 2022 | % * | Net LLPs ** | EBTP 31 December 2022 | % *** | LLAs 31 December 2022 | % **** |
|---|---|---|---|---|---|---|---|
| CGD | 102,503,009 | 33.44% | −5300 | 1,124,898 | 1.10% | 2,254,541 | 2.20% |
| BCP | 89,860,541 | 29.31% | 300,829 | 424,967 | 0.47% | 1,502,373 | 1.67% |
| Santander | 56,166,620 | 18.32% | −11,943 | 841,856 | 1.50% | 946,296 | 1.68% |
| BPI | 38,904,553 | 12.69% | 66,334 | 527,119 | 1.35% | 519,264 | 1.33% |
| Montepio | 19,106,251 | 6.23% | 13,371 | 93,063 | 0.49% | 374,034 | 1.86% |
| Total | 308,977,957 | 100.00% | 363,291 | 5,596,608 | | 5,596,508 | |

\* Total assets of the bank as a percentage of the total assets. ** If negative, it indicates more reversals than LLPs. *** Earnings before taxes and LLPs (EBTP) as a percentage of the bank's total assets. **** LLAs as a percentage of the bank's total assets.

CGD emerges as the leader with the highest asset value, constituting 33.44% of the total sample, followed by BCP, Santander, BPI, and Montepio. Notably, CGD and Santander reported negative net Loan Loss Provisions (LLPs) in 2022, indicating possible improvements in asset quality, changes in accounting policies, or earnings management. The net LLPs show significant variation across the banks, with CGD and Santander displaying negative values, contrasting with the positive values of other banks. The study also analyzes the ratio of earnings before taxes and LLPs over total assets as a metric for earnings management, revealing varying levels of LLPs across the banks, suggesting differences in how they manage results. Additionally, CGD stands out in terms of LLAs, both in absolute terms and relative to total assets, with Montepio showing the lowest absolute LLAs value but a high value relative to its assets, indicating potential risk or overvaluation in its loan portfolio.

Figure 2 analysis over a recent period highlights fluctuating GDP growth in Portugal, starting with an increase in 2013, followed by a minor decline in 2014, and then stabilizing with moderate growth up to 2018. The COVID-19 pandemic in 2020 caused a significant GDP drop with a subsequent sharp recovery. The economic rebound continued into 2021 and 2022, though with a deceleration by the second semester of 2022. LLPs remained stable throughout the period, suggesting a consistent credit risk environment potentially due to legislative measures and the IASB's intervention, which may have mitigated pro-cyclical behavior in LLP recognition during the economic crisis, thereby contributing to financial sector stability.

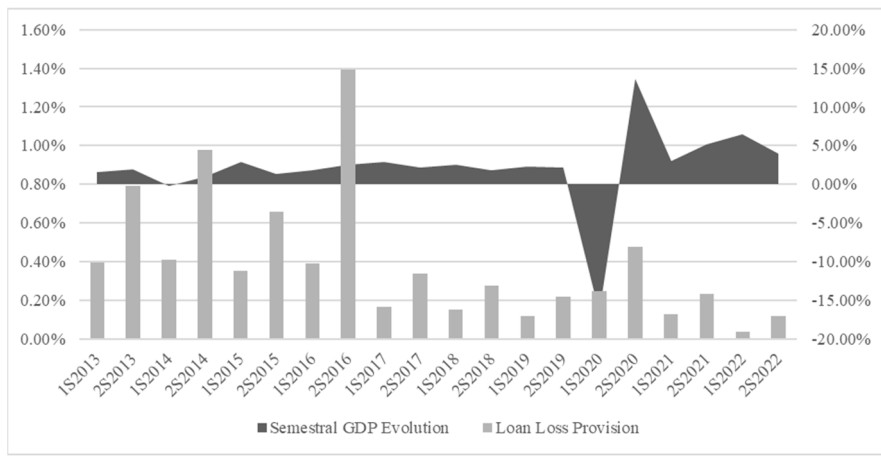

**Figure 2.** LLPs Recognition across economic cycles (2013 to 2022). Semestral GDP Evolution | Net LLP over total assets of banks.

### 4. Results

To test the first hypothesis (H1: The recognition of LLPs in Portuguese banks, according to the IFRS 9's ECL model, exhibits pro-cyclicality) and the second hypothesis (H2: The recognition of LLPs in Portuguese banks, according to the IFRS 9's ECL model, allows for earnings and equity management), a multiple linear regression model was utilized. This model describes the relationship between the dependent variable LLP (explained variable), the explanatory variables representative of economic cycles, and the management of earnings and equity following the methodology adopted in the studies of Araújo et al. (2018), Beatty and Liao (2014), and Casta et al. (2019).

Before proceeding with the correlation analysis of the variables, their normality was tested to decide between the Pearson and Spearman tests for analyzing the correlations among the various model variables. From the analysis conducted, it was concluded that the variables $EBTP_{it}$, $EQUITY_{it}$, and $LOANS_{it}$ follow a normal distribution, and the remaining variables do not. Therefore, the Spearman correlation matrix was chosen, as presented in Table 4.

**Table 4.** Spearman Correlation.

| | $LLP_{it}$ | $PIB_t$ | $UNEMPLOYED_t$ | $EBTP_{it}$ | $EQUITY_{it}$ | $\Delta LOANS_{it}$ | $LOANS$ | $LLA_{it}$ | $SIZE_{it}$ | $IFRS_t$ |
|---|---|---|---|---|---|---|---|---|---|---|
| $LLP_{it}$ | 1 | | | | | | | | | |
| $PIB_t$ | −0.297 *** | 1 | | | | | | | | |
| $UNEMPLOYED_t$ | −0.122 | 0.350 *** | 1 | | | | | | | |
| $EBTP_{it}$ | −0.124 | 0.030 | −0.164 | 1 | | | | | | |
| $EQUITY_{it}$ | −0.450 *** | 0.273 *** | −0.091 | 0.387 *** | 1 | | | | | |
| $\Delta LOANS_{it}$ | −0.217 ** | 0.260 *** | 0.127 | 0.102 | 0.288 *** | 1 | | | | |
| $LOANS_{it}$ | −0.073 | 0.136 | 0.029 | −0.020 | 0.118 | 0.250 ** | 1 | | | |
| $LLA_{it}$ | 0.595 *** | −0.323 *** | −0.091 | −0.262 *** | −0.323 *** | −0.370 *** | −0.120 | 1 | | |
| $SIZE_{it}$ | −0.019 | 0.041 | 0.034 | 0.084 | −0.035 | −0.002 | −0.398 *** | 0.057 | 1 | |
| $IFRS_t$ | −0.396 *** | 0.434 *** | 0.087 | 0.191 * | 0.548 *** | 0.498 *** | 0.249 ** | −0.581 *** | 0.003 | 1 |

Note: Statistical significance indicated by: *** <1%, ** <5%, and * <10%.

The Spearman correlation matrix in Table 4 reveals several significant associations among the variables used in the model. There is a strong negative correlation between $EQUITY_{it}$ and $LLP_{it}$, suggesting that banks with higher equity tend to recognize fewer LLPs. A strong negative correlation is also observed between Portugal's GDP and the LLPs recognized by the banks in the sample, indicating potential pro-cyclicality in LLP recognition. The moderate positive correlations with control variables $\Delta LOANS_{it}$ and $LOANS_{it}$ were expected, as the data in these variables are related. The $IFRS_t$ variable shows correlations with various variables, except for $UNEMPLOYED_t$ and $SIZE_{it}$. In

summary, the results from the Spearman test provide important preliminary trends and have aided in the selection of variables for the regression model.

Although these correlations are an important basis for model construction, it is essential to perform a more thorough analysis when using a multiple linear regression, which requires verifying the assumptions that validate the model used (Laureano 2020, p. 236). To analyze the presence of autocorrelation among the regression residuals, the Durbin-Watson test was conducted, yielding a result of 1.52, which indicates that the assumption of error independence is verified. The normal distribution of errors was tested using the Kolmogorov-Smirnov normal distribution adherence test, with the null hypothesis of error normality being rejected ($p$-value < 0.01). However, in large samples ($n > 30$), as in this study ($n \geq 100$), regression results can be considered in a linear regression model (Laureano 2020). Multicollinearity was tested in all variables, and the assumption of its absence was verified. Additionally, the F-test (ANOVA) was conducted to verify the global significance of the model, allowing for statistical inference. This test confirmed that the model is adequate for explaining the relationship between the dependent variable and the independent variables, being statistically significant ($p$-value < 0.05).

The decision to utilize two sub-models in this study stems from an effort to capture the complex dynamics and significant changes in the regulatory and economic environment, and their impact on the provisioning and reserve practices of Portuguese banks. This approach not only enhances the accuracy of the analysis but also contributes to a deeper understanding of the interactions between accounting policies, macroeconomic factors, and credit risk management. For instance, during the period of the new ECL model, due to the COVID-19 pandemic and rising inflation, regulatory intervention measures were applied that had never been enacted under the previous model. Therefore, it will be crucial to divide the overall model into two sub-models for sub-periods corresponding to the use of the ICL model (2013–2017) and the ECL model (2018–2022). In Table 5, the results of the model estimation are presented for the entire sample (2013–2022) and for the sub-periods corresponding to the use of the ICL (2013–2017) and ECL (2018–2022) models.

**Table 5.** Regression coefficients estimation.

| Variables | Subperiod Models | | |
|---|---|---|---|
| | **Main Model (2013–2022)** | **ICL Model (2013–2017)** | **ECL Model (2018–2022)** |
| H1: | | | |
| $PIB_t$ | (−0.019) | (−0.296) | (−1.510) |
| $UNEMPLOYED_t$ | (0.026) | *** (−3.183) | *** (2.970) |
| H2: | | | |
| $EBTP_{it}$ | *** (2.894) | ** (2.090) | * (1.740) |
| $EQUITY_{it}$ | *** (−4.558) | *** (−4.723) | * (−1.910) |
| Control: | | | |
| $\Delta LOANS_{it}$ | (−1.025) | * (1.92) | (1.110) |
| $LOANS_{it}$ | (1.341) | (1.124) | * (−1.710) |
| $LLA_{it}$ | *** (6.194) | *** (3.846) | ** (2.150) |
| $SIZE_{it}$ | (−0.819) | * (−1.697) | ** (−2.28) |
| $IFRS_t$ | (1.017) | | |
| $R^2$ | 0.502 | 0.625 | 0.447 |
| Teste F | <0.001 | <0.001 | <0.001 |
| N | 100 | 50 | 50 |

Notes: Statistical significance at: *** <1%, ** <5%, and * <10%. T-statistics are in parentheses.

An R² above 50% is a good indicator for the model's performance and acceptance, as well as the F-test (ANOVA) for the global significance of the model (Laureano 2020). In the main model, a determination coefficient (R²) of 0.502 was observed, indicating that 50.2% of the variation in the dependent variable was explained by the independent variables.

Regarding the F-test, both the main model and the subperiod models are adequate for explaining the relationship between the dependent and independent variables, being statistically significant ($p$-value < 0.05).

Among the variables studied in the main model, $EBTP_{it}$ and $EQUITY_{it}$ are statistically significant, with $EBTP_{it}$ showing a positive relationship and $EQUITY_{it}$ a negative relationship with the dependent variable, both behaviors being expected. On the other hand, $PIB_t$, $UNEMPLOYED_t$, $\Delta LOANS_{it}$, $LOANS_{it}$, $SIZE_{it}$, and $IFRS_t$ do not show statistical significance, indicating that they do not have a statistically significant impact on the dependent variable $LLP_{it}$. The control variable $LLA_{it}$ proved to be statistically significant, showing a positive influence on the dependent variable $LLP_{it}$, as expected (Agénor and Zilberman 2015). With the subperiod models, an attempt was made to separately analyze the impact on the dependent variable over two distinct periods: the ICL model (2013 to 2017) and the ECL model (2018 to 2022). In the ICL model, the variables $UNEMPLOYED_t$, $EBTP_{it}$, $EQUITY_{it}$, $\Delta LOANS_{it}$, $LLA_{it}$, and $SIZE_{it}$ had statistically significant impacts on the dependent variable $LLP_{it}$, with the variables $EQUITY_{it}$, $UNEMPLOYED_t$, and $SIZE_{it}$ showing a negative relationship and the other statistically significant variables showing a positive relationship. In the ECL model, the variables $UNEMPLOYED_t$, $EBTP_{it}$, $EQUITY_{it}$, $LOANS_{it}$, $LLA_{it}$, and $SIZE_{it}$ have statistically significant impacts on the dependent variable $LLP_{it}$. The variable $UNEMPLOYED_t$, $EBTP_{it}$, and $LLA_{it}$ show a positive relationship and the other statistically significant variables show a negative relationship with the dependent variable $LLP_{it}$. It is noteworthy that the variable $UNEMPLOYED_t$ is statistically significant in both models, which is not the case in the main model, although with opposite signs.

## 5. Discussion

According to the results presented in the subperiod models, it is possible to validate the first hypothesis with the variable $UNEMPLOYED_t$ (H1: The recognition of LLPs in Portuguese banks, according to the IFRS 9's ECL model, exhibits pro-cyclicality) showing, in the subperiod of the ECL model, a positive relationship with the dependent variable $LLP_{it}$, meeting the initial expectations that a dynamic model reacts positively to individual economic conditions (unemployment), recognizing more LLP (Chan-Lau 2012; Novotny-Farkas 2016; Wezel 2010).

Although the variable $PIB_t$ is not statistically significant for the ECL model ($p$-value = 0.138), it is closer to being significant compared to the ICL model ($p$-value = 0.769). This is typical of a dynamic model, where LLP recognition occurs not at the moment of default, but when economic conditions show negative signs (Bushman and Williams 2015; Casta et al. 2019; Gebhardt and Novotny-Farkas 2011; Laux 2012). Thus, regarding the first hypothesis (H1), the expected pro-cyclicality was evidenced, as reviewed in various studies, for the Portuguese context (Agénor and Zilberman 2015; Bikker and Metzemakers 2005; Bouvatier and Lepetit 2012; Borio et al. 2001; Chen et al. 2020; Gomaa et al. 2019; Morrison and White 2010; Novotny-Farkas 2016; Olszak et al. 2017; Ozili and Outa 2017; Pastiranová and Witzany 2021; Pucci 2017). This result is consistent with some studies suggesting potential difficulties for managers in incorporating indicators and implementing countercyclical reserves (Du et al. 2023; Seitz et al. 2018), as well as failing to reveal economic factors, as happened during the COVID-19 pandemic (Barnoussi et al. 2020). These results may also be influenced by the specific behavior of Portuguese banks, which, due to the history of financial crises (the most recent in 2011 with a new financial bailout), have become more restrained in recognizing LLP. Furthermore, ongoing assistance from the Portuguese state to banks, businesses, and families (e.g., state guarantees in moratoriums granted during the COVID-19 pandemic) may have delayed the emergence of defaults, or increased credit risk. Another possible explanation is related to the restraint in recognizing LLP during the COVID-19 pandemic and regulators' concerns about banks' regulatory capital (Barnoussi et al. 2020; Bischof et al. 2021). This conservative approach to credit risk, already signaled, could result in delays in recognizing LLP (EBA 2021).

Regarding Hypothesis 2 (H2: The recognition of LLPs in Portuguese banks, according to the IFRS 9's ECL model, allows for earnings and equity management), evidence of a significant relationship for the two explanatory variables ($EBTP_{it}$ and $EQUITY_{it}$) in the ECL model was also found, meeting the initial expectations (Du et al. 2023; Gomaa et al. 2019; Novotny-Farkas 2016; Ozili and Outa 2017). However, this statistical relevance is stronger in the ICL model ($p$-value < 0.05) than in the new ECL model ($p$-value < 0.10). This more statistically significant relevance in the previous ICL model, regarding the management of earnings and equity by Portuguese banks in the recognition of LLP, aligns with studies reporting delays in recognizing LLP in the ICL model of IAS 39 and even hiding inevitable losses (Gebhardt and Novotny-Farkas 2011; Hoogervorst 2014). The unified European regulation, through the EBA, may have minimized the impact of LLP recognition on earnings management in the ECL model compared to ICL (Nnadi et al. 2023), thereby demonstrating that regulation is important in minimizing the possibility of earnings management, providing discipline and transparency to the market (Onali et al. 2021).

This article also confirmed the positive influence of LLAs on LLPs recognition in both models, demonstrating that LLPs are recognized considering existing LLAs. This result corroborates the empirical evidence previously found in the literature review, that the positive relationship between LLAs and LLPs recognition (Agénor and Zilberman 2015) reduces the ability of bank managers to increase loans to existing risk clients, in both ECL and ICL models (Du et al. 2023).

Regarding the size of banks, it was expected that it would positively affect LLP recognition—the larger the bank, the higher the level of LLP recognition (Mechelli and Cimini 2021; Onali et al. 2021). However, the study's results indicate that, with the ECL model of IFRS 9, the size of Portuguese banks negatively affects LLP recognition. This intriguing result in the context of the Portuguese reality diverges from the reviewed literature. Containment in recognizing LLP, as analyzed, can be considered one of the possible causes for this result. Moreover, it is important to note that three of the four largest banks in the sample underwent public recapitalization through the issuance of contingent convertible bonds (CoCos) in June 2012 (BCP, BPI, and CGD), which may have negatively impacted LLP recognition in subsequent years.

## 6. Conclusions

This article represents a pioneering approach in analyzing the pro-cyclicality of LLPs in Portuguese banks, as well as the management of earnings and equity through their recognition. The financial sector is crucial for any country, especially in a scenario of rising interest rates, potentially playing a significant role in managing an impending economic and/or financial crisis. In Portugal, a country that has recently sought external assistance, the financial sector gains particular relevance since, in an overly leveraged economy, the adequacy of banks' reserves is fundamental to overcoming an economic recession scenario.

This article study aimed to analyze the pro-cyclicality of LLPs in Portuguese banks, as well as the existence of earnings and equity management through their recognition, over a time frame from 2013 to 2022. The study focused on 5 Portuguese commercial banks, selected from a base of 17 banks from the Portuguese Banking Association, excluding 12 due to the lack of data. A multiple linear regression model was used, identifying how LLPs react to the behavior of explanatory variables, economic cycles, and the management of earnings and equity, over an extended period of 10 years (2013 to 2022), following the methodology adopted by Araújo et al. (2018), Beatty and Liao (2014), and Casta et al. (2019).

The results confirm the pro-cyclicality in the recognition of LLPs in Portuguese banks, in line with the first hypothesis (H1: The recognition of LLPs in Portuguese banks, according to the IFRS 9's ECL model, exhibits pro-cyclicality), where the ECL model of IFRS 9 responds positively to economic indicators like unemployment, even though GDP did not show statistical significance. This pro-cyclicality reflects the ECL model's reaction to adverse economic contexts, corroborating the literature in the Portuguese context.

Regarding the second hypothesis (H2: The recognition of LLPs in Portuguese banks, according to the IFRS 9's ECL model, allows for earnings and equity management), a significant relationship in the management of earnings and equity in the ECL model is verified, although less pronounced compared to the previous ICL model. This evidence suggests that despite the EBA regulation and the new accounting model potentially attenuating earnings management, it continues to be a practice in LLP recognition, albeit less intensively, highlighting the importance of regulation to ensure discipline and transparency in the financial market.

This article makes significant contributions to the literature on understanding the impact of the ECL model of IFRS 9 on Portuguese banks. The results validate the hypothesis of pro-cyclicality in the ECL model of IFRS 9, showing that it is more sensitive to economic conditions compared to the ICL model of IAS 39, confirming expectations from Abad and Suarez (2017). However, given the current macroeconomic conditions, such as the conflict in Ukraine, inflation, rising interest rates, and weak growth in Europe, the ECL model of IFRS 9 seems not to be anticipating the recognition of LLPs, suggesting it may not be performing as expected by Portuguese banks, i.e., creating adequate countercyclical reserves (LLAs) to face future crises. One possible cause for this could be the systematic intervention of the state, which is not creating incentives for Portuguese banks to reflect macroeconomic conditions ($PIB_t$), focusing only on microeconomic conditions ($UNEMPLOYED_t$), i.e., financing efforts and defaults of families and businesses. These events should have influenced the increase in reserves in a timely manner, which did not happen, potentially leading to restraint in recognizing LLPs. This study also brings contributions to practice, particularly for regulators and professionals in the field, as the expected increase in recognizing LLAs, anticipating adverse economic conditions, was not observed, alerting to the existence of earnings and equity management by Portuguese banks. However, earnings and equity management are less pronounced in the ECL model compared to the old ICL model, indicating that greater transparency and better European regulatory mechanisms by the EBA may have attenuated this behavior.

Like any scientific study, this one also has limitations. Firstly, the study focused solely on Portuguese banks, limiting the generalization of the results to other jurisdictions. Moreover, the sample does not cover all financial institutions in Portugal due to the limitations mentioned in the sample definition, being a reduced sample of banks, a consequence of the intention to study in depth a single jurisdiction, Portugal. Another limitation refers to the analysis period, which may not encompass all relevant economic variations over time. Lastly, the study focused on quantitative analysis, leaving out possible qualitative explanations that could enrich the understanding of the phenomena under study.

In future studies, it would be important to analyze and compare the impact of government policies and financial assistance on the pro-cyclicality in recognizing LLPs in the ECL model. Additionally, it is relevant to investigate whether the reserves of the ECL model in Portuguese banks are adequate to face a financial crisis similar to the last high-risk mortgage credit crisis (subprime). Although the study points to an anticipation in recognizing LLPs, they may not be sufficient for a large-scale crisis. Another interesting line of research would be to study the behavior of the ECL model over a more extended period, using estimated models developed in some studies on this area. Finally, it would be important to estimate the impact of LLPs on Portuguese banks' equity and regulatory capital if there had been no government assistance and recommendations, like the one from IASB (2020), for restraint in recognizing LLPs.

**Author Contributions:** Conceptualization, M.R.; methodology, M.R.; validation, C.C. (Carla Carvalho) and C.C. (Cecília Carmo); formal analysis, M.R.; investigation, M.R.; writing—original draft preparation, M.R.; writing—review and editing, C.C. (Carla Carvalho) and C.C. (Cecília Carmo) All authors have read and agreed to the published version of the manuscript.

**Funding:** This research received no external funding.

**Data Availability Statement:** Publicly available datasets were analyzed in this study with sources as outlined in Section 3.2.

**Conflicts of Interest:** The authors declare no conflicts of interest.

## Notes

[1] In the specialized literature on the subject, it is common to encounter the terms Loan Loss Provisions (LLPs) and Loan Loss Allowances (LLAs) to designate credit impairment losses. For the sake of uniformity in the terminology used in studies, this research will use the abbreviation LLP to express credit impairment losses recognized in the period, and LLA for the accumulated credit impairment losses, following the approach of Salazar et al. (2023).

[2] In this study, the term earnings management is used to describe deliberate intervention by managers through the recognition of LLP and its impact on the banks' results and equity. According to Schipper (1989), this manipulation is an intentional intervention in financial information to obtain specific benefits, particularly through the selection of accounting practices that align more closely with the interests of managers or the company. Similarly, Healy and Wahlen (1999) note that earnings management occurs when managers use their judgment to alter financial reports to influence stakeholders' perception or meet specific contractual clauses.

[3] During the COVID-19 pandemic, the IASB intervened, limiting the recognition and increase of LLPs for phase 2 (significant increase in risk) through a communication that altered the accounting policy of the ECL model. In this note, the IASB indicates that the impact on loans due to moratoriums, when backed by the states, should not be interpreted as a significant increase in risk.

[4] "LLA = EAD × PD × LGD".

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
