# Peer review of "Impacts of the Expected Credit Loss Model on Pro-Cyclicality, Earnings Management, and Equity Management in the Portuguese Banking Sector"

_jrfm, doi:10.3390/jrfm17030112_

Round 1

Reviewer 1 Report

Comments and Suggestions for Authors

#1 The author conducts empirical analysis on the topic of "Impacts of the Expected Loss Model on Pro-Cyclicality, Earnings Management, and Equity Management." The study aims to explore the lack of practical applicability in the relevant content based on empirical evidence. In other words, the author must articulate more clearly the distinctions between the content of this research topic and the findings of previous scholars.

#2 The construction of the research hypothesis lacks a comprehensive framework presentation, and it fails to provide objectivity and effectively represent the significance and reference value of the research problem.

#3
During the annual period under investigation, which corresponds to the year of COVID-19, the author examines whether the relevant samples exhibit significantly greater differences compared to other periods. In order to generate a more stable empirical analysis and reduce bias in hypothesis testing, the data within this financial system should be compared to long-term averages. This comparative approach enables a more robust assessment of the empirical analysis within the context of the system, mitigating potential biases in hypothesis verification.

Comments on the Quality of English Language

Minor editing of English language required.

Author Response

We appreciate the feedback from the reviewers. In an effort to reconcile the two reviews, I present the following notes on the modifications made based on your suggestions:

  1. Modification of the title to clarify the target audience and the scope of the article.
  2. Revision of the fifth paragraph to more robustly substantiate the study's objective and its contribution to the existing literature in the field.
  3. Correction made in subsection 2.1.2.
  4. Elimination of subsection 2.4, "hypothesis formulation", integrating the hypotheses into the context of the literature review for improved flow and cohesion in reading.
  5. This article is partially based on a doctoral thesis. Initially, I included the entire content in the article. However, in order to reduce the length of the article, I chose to exclude paragraphs related to agency and institutional theories, as well as associated references. Similarly, I simplified the statistical analysis section by removing tables 4 and 5. The analysis of table 3 and figure 1 was condensed. Regarding subsection 4.2, "analysis and discussion of results", I decided to divide the content into two distinct chapters: chapter 4, "results", and chapter 5, "discussion". I believe these changes will make the text more accessible and aligned with the standards of scientific articles published in this journal.

Reviewer 2 Report

Comments and Suggestions for Authors

This paper explores an intriguing topic of the relationship between Loan Loss Provisions (LLP) and Earnings Management (EM) in the context of IFRS 9, with a focus on Portugal. While the paper boasts several strengths, there are key areas that could be improved to enhance its suitability for journal publication. I limit my comments to more overall aspects:

  1. Readability: The manuscript currently challenges the reader with its length and complexity. A reduction of at least 30% in length would align it more closely with standard journal article formats. A more focused approach with less extensive elaboration is recommended. The current style suggests this is a transition from a master's thesis to a journal-like format (e.g. see the word "Modelo" in Table 5). Moreover, readers generally prefer a more concise and accessible article.

  2. Title Clarity: The current title does not effectively convey the paper's content, which could impede understanding and future citations. A more direct and clear title would be beneficial.

  3. Contribution: The contribution of this paper, though modest, is acknowledged by the authors. It's commendable that they have addressed this aspect, but further emphasis on the paper's unique contributions could enhance its impact.

  4. Justification of Empirical Setting: While there are mentions of why Portugal is an appropriate setting for this study, these justifications could be strengthened. Typically, the broader topic is introduced before delving into the specifics of the empirical setting. In this paper, the focus on Portugal is immediate and pronounced, even in the hypotheses, which may give the impression of a master's level analysis. Reframing this section to first present the general topic and subsequently justify the choice of Portugal would improve the paper's academic rigor.

  5. Table texts - are to some extent too limited. One should always be able to read the Table without the need for consulting the text in the manuscript.
Comments on the Quality of English Language

The language is OK, but there are minor errors.

Author Response

(The authors gave the same response as above.)

Round 2

Reviewer 2 Report

Comments and Suggestions for Authors

I have reviewed the updated version of the manuscript. In my opinion, it has become more readable and represents an acceptable piece of work. However, the manuscript has been condensed to a standard format suitable for journal submission, yet this restructuring lacks finesse. The paper would benefit from a more rigorous structure and tighter focus.

Why is it essential to emphasize Portugal in the title?

Author Response

Thank you very much for your constructive feedback. In response to your suggestions for a more rigorous structure and a tighter focus, I have implemented changes aimed at improve the manuscript.

Change 1: Streamlining Sections on Cyclicality

To address the concern for a more cohesive narrative, I have consolidated the sections dealing with the cyclicality and pro-cyclicality of LLPs. Previously segmented into sub-sections (2.1.1 Overview, 2.1.2 Subsubsection, 2.1.2 Cyclicality and Pro-Cyclicality in the ICL Model of IAS 39, and 2.1.3 Cyclicality and Pro-Cyclicality in the ECL Model of IFRS 9), these have now been merged into a single, focused section: 2.1. Cyclicality and Pro-Cyclicality of LLPs. This restructuring allows for a more fluid and focused discussion, particularly in the initial paragraphs of subsections 2.1.2 and 2.1.3, thereby enhancing the coherence and the depth of the analysis on the cyclicality inherent in both models.

Change 2: Consolidating on 2.3. Impact of Dynamic Models on Economic Cycles Management

Similarly, to further harmonize the manuscript's structure and ensure a seamless narrative flow, I merged sections 2.3. Impact of Dynamic Models on Economic Cycles Management, 2.3.1. Dynamic Models of LLPs, and 2.3.2. Analysis of the IFRS 9 ECL Model into a single, comprehensive section. This revision not only improves readability but also strengthens the connection with the study's second hypothesis, maintaining the integrity and relevance of all previously cited articles.

Change 3: Emphasizing Portugal in the Title

Regarding the importance of emphasizing Portugal in the title, this decision was made to effectively convey the article's content and scope, facilitating future comprehension and citations. The title "Impacts of the Expected Credit Loss Model on Pro-Cyclicality, Earnings Management, and Equity Management in the Portuguese Banking Sector" is intended to be clear and direct, signaling to prospective readers the specific focus and geographical context of the study. This approach ensures that the research is immediately relevant to scholars and practitioners interested in the implications of IFRS 9 within the Portuguese banking sector, while also contributing to the broader discourse on accounting standards and financial regulation.

I hope these revisions meet your expectations and address the concerns raised in your review. I am committed to ensuring the manuscript achieves the highest standards of academic rigor and relevance, and I am open to further suggestions or modifications you deem necessary.

Thank you once again for your valuable feedback.

Best regards
